Temperature-induced changes in egg white antimicrobial concentrations during pre-incubation do not influence bacterial trans-shell penetration but do affect hatchling phenotype in Mallards

Svobodová Jana 1
Kreisinger Jakub 2
Gvoždíková Javůrková Veronika veronika.javurkova@gmail.com 3 4
1 Faculty of Environmental Sciences, Department of Ecology, Czech University of Life Sciences , Prague , Suchdol , Czech Republic
2 Faculty of Science, Department of Zoology, Charles University Prague , Prague , Czech Republic
3 Institute of Vertebrate Biology of the Czech Academy of Sciences , Brno , Czech Republic
4 Faculty of Agrobiology, Food and Natural Resources, Department of Animal Science, Czech University of Life Sciences , Prague , Suchdol , Czech Republic
Azevedo Vasco
Electronic publication date: 2021 Nov 11
Publication date: 2021
Volume: 9
Electronic Location ID: e12401
Received 2021 May 10; Accepted 2021 Oct 6
Copyright: ©2021 Svobodová et al.
Copyright year: 2021
Copyright holder: Svobodová et al.
License: This is an open access article distributed under the terms of the Creative Commons Attribution License, which permits unrestricted use, distribution, reproduction and adaptation in any medium and for any purpose provided that it is properly attributed. For attribution, the original author(s), title, publication source (PeerJ) and either DOI or URL of the article must be cited.
License URL: https://creativecommons.org/licenses/by/4.0/

Keywords: Albumen, Antimicrobial proteins, Embryo viability, Microorganisms, Bacterial penetration, Incubation, Temperature

Funding: Czech Academy of Sciences RVO: 68081766 This study was supported through institutional research support of the Czech Academy of Sciences, RVO: 68081766. The funders had no role in study design, data collection and analysis, decision to publish, or preparation of the manuscript.

==============================
Microbiome formation and assemblage are essential processes influencing proper embryonal and early-life development in neonates. In birds, transmission of microbes from the outer environment into the egg’s interior has been found to shape embryo viability and hatchling phenotype. However, microbial transmission may be affected by egg-white antimicrobial proteins (AMPs), whose concentration and antimicrobial action are temperature-modulated. As both partial incubation and clutch covering with nest-lining feathers during the pre-incubation period can significantly alter temperature conditions acting on eggs, we experimentally investigated the effects of these behavioural mechanisms on concentrations of both the primary and most abundant egg-white AMPs (lysozyme and avidin) using mallard (Anas platyrhychos) eggs. In addition, we assessed whether concentrations of egg-white AMPs altered the probability and intensity of bacterial trans-shell penetration, thereby affecting hatchling morphological traits in vivo. We observed higher concentrations of lysozyme in partially incubated eggs. Clutch covering with nest-lining feathers had no effect on egg-white AMP concentration and we observed no association between concentration of egg-white lysozyme and avidin with either the probability or intensity of bacterial trans-shell penetration. The higher egg-white lysozyme concentration was associated with decreased scaled body mass index of hatchlings. These outcomes demonstrate that incubation prior to clutch completion in precocial birds can alter concentrations of particular egg-white AMPs, though with no effect on bacterial transmission into the egg in vivo. Furthermore, a higher egg white lysozyme concentration compromised hatchling body condition, suggesting a potential growth-regulating role of lysozyme during embryogenesis in precocial birds.

Introduction

Host-microbiome interactions and assemblage during embryonal and early-life phases appear to be strong determinants of the prosperity and overall success of progeny (Calatayud, Koren & Collado, 2019; Campos-Cerda & Bohannon, 2020; Chen et al., 2020; Osorio, 2020; Voirol et al., 2020).

Avian eggshell microbiomes are diverse (Grizard et al., 2014; Van Veelen, Salles & Tieleman, 2018) and are primarily shaped by the nest material, the local environment and the parent’s skin and feathers (Diaz-Lora et al., 2019; Martinez-Garcia et al., 2016; Ruiz-Castellano et al., 2016; Van Veelen, Salles & Tieleman, 2017). As documented under both natural (Cook et al., 2003) and experimental conditions (Javůrková et al., 2014; Wang et al., 2018), these eggshell microorganisms are capable of entering avian egg content, a process known as microbial trans-shell penetration. Unlike the broadly documented associations between eggshell microbiota and hatching success (e.g., Lee et al., 2017; Peralta-Sanchez et al., 2018), studies investigating proximate effects of penetrating microorganisms on avian embryos and hatchlings are scarce, with the few published reports to date documenting suppressed embryo viability (Cook et al., 2005a; Cook et al., 2003; Cook et al., 2005b; Wang & Beissinger, 2011) or decreased residual body weight of hatchlings (Javůrková et al., 2014).

Birds have evolved numerous egg-related and behavioural mechanisms to protect against uncontrolled proliferation of microbes outside and inside the egg. Eggshell pigmentation (Ishikawa et al., 2010), eggshell microstructure characteristics (D’Alba et al., 2017; Grellet-Tinner, Lindsay & Thompson, 2017; Martin-Vivaldi et al., 2014), cuticle nanostructuring (D’Alba et al., 2014), deposition of antimicrobial proteins into eggshell structures (Bain et al., 2013; Gautron et al., 2011; Wellman-Labadie, Picman & Hincke, 2008a) and nest material (Diaz-Lora et al., 2019; Ruiz-Castellano et al., 2019; Ruiz-Castellano et al., 2016) have all been found to significantly shape eggshell microbiota. In comparison, mechanisms reducing microbial trans-shell penetration and proliferation of microbes inside the egg have primarily been linked with egg incubation (Fang et al., 2012a; Svobodová et al, 2019) and the concentration of miscellaneous egg-white proteins (Mann & Mann, 2011; Sun et al., 2017), especially those having antimicrobial potential (Baron & Jan, 2011; Svobodová et al, 2019; Wellman-Labadie, Picman & Hincke, 2008b).

Among the most abundant, and most investigated, egg-white antimicrobial proteins (AMPs) are lysozyme, ovotransferrin and avidin (Ibrahim, 2019; Sun et al., 2017). While lysozyme shows strong bactericidal activity against both G+ and G- bacteria (Ibrahim, Matsuzaki & Aoki, 2001; Wellman-Labadie, Picman & Hincke, 2008b), avidin and ovotransferrin are more bacteriostatic (Guyot et al., 2016a; Guyot et al., 2016b; Wellman-Labadie, Picman & Hincke, 2008b) due to their ability to reversibly bind biotin and iron, thus making them unavailable for bacterial growth (Board & Fuller, 1974; Pierce et al., 2021; Wu & Acero-Lopez, 2012). Owing to their broad antimicrobial potential, most egg-white AMPs are considered as essential maternal effects transferred into the egg in birds (Bonisoli-Alquati et al., 2010; Saino et al., 2007). For example, higher egg-white lysozyme concentration was found in eggs produced after mating with a better quality male (Saino et al., 2007), or to be associated with improved hatchability and innate immunity of hatchlings (Saino et al., 2002). Egg-white proteomic profiles may change significantly during the incubation cycle, however, with increases in the concentration and relative abundance of egg-white AMPs due to the water loss (Guyot et al., 2016b), phosphorylation, glycosylation or decreased solid albumin (Zhu et al., 2019), and decreases in concentration due to the formation of protein complexes and/or protein aggregations (Liu, Qiu & Ma, 2015; Qiu et al., 2012) during the first 12 days of incubation, all of which may lead to alterations in egg-white antimicrobial potential (Fang et al., 2012a; Fang et al., 2012b; Grizard et al., 2015). Further, the different incubation patterns of altricial and precocial birds have been shown to result in temperature-induced changes in egg-white lysozyme and ovotransferrin concentration, enhancing the proliferation of beneficial probiotic microorganisms in the egg-white (Svobodová et al, 2019). It is likely, therefore, that while incubation may significantly shift eggshell microbiota (Bollinger et al., 2018; Grizard et al., 2014; Grizard et al., 2015; Ruiz-De-Castaneda et al., 2011; Ruiz-de Castaneda et al., 2012), its selective antimicrobial effect inside the egg is most probably inherent in mediation of changes in the egg-white chemical and proteomic profile, where an increase/decrease in lysozyme and avidin would be expected. Furthermore, as evidence exists for a physiological role of lysozyme and avidin on the developing embryo, resulting in alterations of body morphological traits of hatchlings (Javůrková et al., 2015; Krkavcová et al., 2018), incubation-mediated changes in egg-white AMP concentration may also significantly shape the hatchling’s morphology. To date, however, experimental evidence for the interactive effects of egg-white AMP profile and incubation under natural conditions are lacking.

Another behavioural mechanism with the potential to reduce risks of microbial trans-shell penetration is clutch covering with nest-lining material during the pre-incubation period. Recent studies suggest that the nest material and nest-lining feathers most likely affect the eggshell microbial assemblage via antimicrobial agents produced by microorganisms in the preen gland (Diaz-Lora et al., 2019; Ruiz-Castellano et al., 2019; Ruiz-Castellano et al., 2016). In addition to the direct antimicrobial action of nest-lining feathers, clutch covering protects the exposed clutch against ambient temperature fluctuations during the pre-incubation period (Pinowski et al., 2006; Prokop & Trnka, 2011). While clutch covering with nest-lining material appeared to have no effect on bacterial trans-shell penetration in a previous experimental study (Javůrková et al., 2014), it may affect the temperature acting on exposed eggs, thereby inducing temperature-mediated changes in egg-white AMP concentration. To date, however, the proximate role of clutch covering with nest-lining feathers on egg-white AMP concentration has not yet been evaluated.

In this study, we experimentally test whether partial incubation and clutch covering with nest-lining feathers during pre-incubation affects concentrations of the two principle egg-white AMPs, lysozyme and avidin, using precocial mallard eggs exposed in their natural breeding habitat. Partial incubation, a behaviour preceding full incubation of the complete clutch observed in many bird species (Wang, Firestone & Beissinger, 2011), keeps the eggs dry (D’Alba, Oborn & Shawkey, 2010), modulates eggshell microbiota (Bollinger et al., 2018; Cook et al., 2005a), may also have an antipredator function (Kreisinger & Albrecht, 2008; Morosinotto, Thomson & Korpimaki, 2013) and could play a role in regulating hatching asynchrony (Magrath, 1990). Whether partial incubation affects or stabilises the egg content antimicrobial properties remains unknown. Furthermore, as an increase in egg-white AMP concentration enhanced in vitro egg-white antimicrobial activity against selected bacterial strains in our previous experimental study (Svobodová et al, 2019), we hypothesise that different concentrations of egg-white AMPs will affect the probability and intensity of bacterial trans-shell penetration in vivo. Finally, as egg-white lysozyme and avidin have been shown to play a role in modulating hatching success, immune response and body morphological traits of hatchlings in other precocial birds (Bonisoli-Alquati et al., 2010; Cucco et al., 2007; Javůrková et al., 2015; Krkavcová et al., 2018), we predict that egg-white lysozyme and avidin concentration will play a similar role in dictating mallard duckling morphological traits.

Material & Methods

Ethical statement

All experiments and analyses were performed in accordance with relevant institutional guidelines and regulations. The experiment was carried out under institutional permission No. 63479/2016-MZE-17214, issued by the Ministry of Agriculture on behalf of the Government of the Czech Republic.

Experimental procedures

Freshly laid mallard eggs were obtained (n = 160) from a commercial hatchery (Mokřiny Duck Farm, Třeboň Fisheries Ltd, Czech Republic) in June 2010. To control for potential bias in the variability of egg-white AMP concentration in experimental eggs due to female identity and egg laying order (D’Alba et al., 2010; Valcu et al., 2019), the experimental eggs were collected randomly at the same time over a single day, thereby ensuring that they came from a similar laying order sequence and from different females. Egg length and width were measured with digital callipers (0.01 mm accuracy; Kinex, Prague, Czech Republic) in order to compute egg volume (Rohwer, 1988). Subsequently, each egg was cleaned with 70% ethanol to eliminate the initial eggshell bacterial assemblage and then placed into sterile portable boxes.

Four randomly selected eggs were placed into each experimental nest (N = 40) distributed in typical mallard breeding habitat (Dívčice, Czech Republic, 49°6′N, 14°18′E) and exposed for nine days, corresponding to the mean mallard egg-laying period observed under natural conditions (Krapu et al., 2004). The eggs in each experimental nest were sorted based on a balanced 2  × 2 factorial design (see Fig. S1), i.e., two eggs were covered with a mixture of nest-lining feathers collected from active mallard nests (see details below), while the other two eggs remained uncovered. Subsequently, two eggs (one covered and one uncovered with nest-lining feathers) were incubated daily in an incubator (OvaEasy 190 Advance, Brinsea Products Inc., Titusville, FL, USA) for periods that mimicked the incubation pattern observed during the mallard pre-incubation period (see Loos & Rohwer, 2004; Javůrková et al., 2014; Table S1). Experimentally incubated eggs were transferred from the experimental nests to the incubator and back each day in portable sterilised boxes. The experimentally incubated eggs were incubated for a total of 45 h at 37.6 °C, with a relative humidity of 60%, over the nine days. The two remaining eggs from each experimental nest were not incubated, but were turned and handled daily to maintain a manipulation procedure identical to that of the eggs transferred for incubation. All experimental eggs were turned 180° twice daily using rubber gloves to maintain optimal egg hatchability (Oliveira et al., 2020).

The nest-lining feathers used for egg covering were collected from several (n = 30), randomly chosen, active mallard nests located within the Dívčice breeding habitat over May 2010. No more than 40–50% of nest-lining material was ever taken from the active mallard nests to prevent nest desertion and reproductive failure of the breeding female. The nest-lining feathers were mixed before being used for the experimental treatment (i.e., covering of experimental eggs by nest-lining feathers; see also Javůrková et al., 2014 for details).

Egg-white sampling and assessment of egg hatchability

Egg-white sampling procedures were identical to those described in detail in Javůrková et al. (2014). In short, all experimental eggs were cleaned with 70% ethanol and the shell gently perforated with a 22 G (0.7 × 40 mm) sterile needle (Terumo®, Germany) at the blunt end. Thereafter, 300 µL of egg-white was removed with a 0.5-mL sterile syringe (B Braun, Germany) and placed in sterile cryotubes stored at −20 °C until egg-white AMP analysis. Needle perforations in the eggshell were sealed using a gel-based adhesive (Loctite Super Attack, Henkel, USA). Based on previous studies (Bonisoli-Alquati et al., 2007; Finkler, Van Orman & Sotherland, 1998), such a procedure has no significant impact on egg hatchability.

The eggs were then placed back in an incubator with temperature at 37.6 °C and relative humidity at 60%, with relative humidity being increased to 80% during the egg-hatching period in order to achieve optimal hatching conditions (Stubblefield & Toll, 1993). The weight (±0.1 g) and tarsus length (±0.1 mm) of each duckling was measured immediately after hatching in order to obtain body mass condition indices (i.e., residual body mass and scaled body mass index (BMI), see below for details).

Analysis of egg white AMP concentration

Concentration of egg-white lysozyme

Lysozyme concentration (mg/mL) was measured using the agar well-diffusion assay of Osserman & Lawlor (1966), which is described in detail in our previous studies (Javůrková et al., 2015; Svobodová et al, 2019). Briefly, 50 mg of lyophilised Micrococcus lysodeikticus (Sigma–Aldrich, ATTC 4698, M3770) was diluted in 10 mL of Britton–Robinson buffer (pH 7.0) prepared via adding 0.492 g boric acid, 0.782 g 98% phosphoric acid, 0.480 g acetic acid and 0.840 g NaOH into 305 ml of distilled water. This suspension was added to a 60 °C solution of 1% agar (1 g of agar (Alchimica) re-suspended in 100 ml of Britton–Robinson buffer) and poured into Petri dishes and left for 30 min to solidify. Core borer was used to punch three mm diameter holes into the agar. Then, homogenisation of each egg-white sample in a glass vial at 1,800 rpm for 15 min was conducted using an immersed cross spin magnetic stirrer bar (12  × 30 mm) and a magnetic stirrer (RH Digital, IKA, Oxford, UK). Subsequently, 10 µL of each egg-white sample was transferred into the holes on the agar plates in duplicate. Standard solutions (10 µL) of lyophilised hen egg-white lysozyme (62971, Fluka) of known concentration (20, 15, 7, 4, 2, 0.5 mg/mL) were also added to the punched holes in each agar plate. The plates were then incubated for 24 h at 21 °C and 50–60% humidity. Diameters of clearance zones around the holes were analysed from photographs using ImageJ. GraphPad Prism version 6.00 for Windows (GraphPad Software, San Diego California USA) was used for interpolation of lysozyme concentration (mg/mL) in each egg-white sample from a calibration curve.

Concentration of egg-white avidin

Avidin concentration (µg/mL) was based on a slightly modified version of the 96-well plate method of Gan & Marquardt (1999) and Shawkey et al. (2008) for assessing the affinity of avidin to biotin, which we used in our previous work (Krkavcová et al., 2018). Briefly, we diluted each egg-white sample 10-fold in carbonate-bicarbonate buffer (Sigma-Aldrich C3041). Then, 100 µL of carbonate-bicarbonate buffer was added to each well (except the first, fifth and ninth well in each row) along rows one to 11 of a Nunc MaxiSorp® flat-bottom 96-well plate (see also Krkavcová et al., 2018). Then each 10-fold diluted egg-white sample was added in volume of 100 µL to empty wells 1, 5 and 12 in each row making four serial dilutions for each of these samples. Finally, 100 µL of avidin standard solution (2.5–0.002 µg/mL; Sigma Aldrich; A9275) diluted in carbonate-bicarbonate buffer was added to the wells in the bottom row. Accurate pipetting of undiluted and diluted egg-white samples was achieved using GENO-DNA S pipette tips (CS960 9405120, Thermo Fisher Scientific) especially designed for viscous liquids. The 96-well plate sealed with parafilm was then incubated at 4 °C overnight. Then we applied same procedures as was described in Krkavcová et al. (2018). Particularly, the content of the wells was poured out and the plate rinsed three times by adding 200 µL of 0.05% Tween washing buffer (Tween 20/PBS) to each well and shaking for five minutes on an IKA KS 260 basic lab shaker. Non-specific protein sites were blocked by adding 200 µL of blocking buffer (1% solution of bovine serum albumin (Sigma Aldrich) in PBS) to each well three times for 30 s, after which 100 µL of a 1:4000 dilution of biotinylated biotin/HRP (Invitrogen, Thermo Fisher Scientific) in Superblock buffer (0.05% Tween 20/blocking buffer) was added to each well and incubated at room temperature for 25 min. The wells were then washed five times with 200 µL of washing buffer, followed by 30 s. shaking on the lab shaker. Then, 100 µL of TMB Substrate blocking buffer (Sigma Aldrich) was added to each well and the plate was incubated at room temperature for 30 min. Subsequent adding of 100 µL of TMB Substrate Stop Reagent (Sigma Aldrich) to each well and mixing it stopped reaction. Using a TECAN Infinite® 200 PRO UV/Vis microplate reader (Tecan Group, Männedorf, Switzerland), we measured sample absorbance at 450 nm, with each sample analysed in duplicate. GraphPad Prism 5 Software was used for interpolation of avidin concentrations (considering four egg-white serial dilutions) from a standard curve for each plate (inter-assay and intra-assay coefficients of variability were 12.6% and 3.2%, respectively).

Quantitative analysis of bacterial trans-shell penetration

Quantitative measurement of bacterial trans-shell penetration (BTSP) was based on our previously published method (Javůrková et al., 2014). In brief, bacterial genomic DNA was extracted from egg-white samples using the EliGene MTB Isolation Kit (Elisabeth Pharmacon, Brno, Czech Republic). Microbial genomic DNA was then analysed for the incidence and intensity of BTSP using RT-PCR based entirely on the targeting of 16S rRNA using an RT-PCR LightCycler 480 system (Roche, Mannheim, Germany). The LightCycler 480 SYBR Green I Master (Roche) and the universal Eubacteria primer set, including forward primer Uni331 (5′-TCCTACGGGAGGCAGCAGT-3′) and reverse primer Uni797 (5′-GGACTACCAGGGTATCTAATCCTGTT-3′), were used for RT-PCR amplification (Horz et al., 2005). We used our previous approach for construction of calibration curves (Javůrková et al., 2014). Specifically, serial dilutions (101 to 109) of purified genomic Streptococcus bovis DNA with a known number of bacterial cells were used. BTSP intensity was then expressed as number of bacterial cells per one mL of egg-white. Relative incidence of BTSP (i.e., penetrated vs. non-penetrated) was based on the successful amplification of diluted Streptococcus bovis DNA which was a positive control for this assignment.

Amplification conditions protocol of Javůrková et al. (2014) was used for setting amplification conditions. Specifically, the PCR reaction was performed in triplicate on a LightCycler_ 480 Multiwell Plate 384 using a total volume of 10 µL, including 5 µL of LightCycler 480 SYBR Green I Master, 3 µL of PCR H2O (Top-Bio,Czech Republic), and 0.5 µL of each primer at concentrations of 5 µM and 1 µL of DNA template. Reaction conditions for DNA amplification were following: one pre-amplification cycle at 95 °C for 10 min followed by 40 amplification cycles at 95 °C for 10 s, 58 °C for 10 s and 72 °C for 30 s at a ramp rate of 4.8 °C/s (Javůrková et al., 2014) Analysis of product melting was performed to determine specificity of amplification. A melting curve was obtained by slow heating at 2.5 °C/s increments from 65 °C to 95 °C, with fluorescence collection at 0.5 °C intervals (see also Javůrková et al., 2014). Efficiency and slope values for particular RT-PCR runs (n = 3) were: 1.82 and 3.84; 2.14 and 3.03; and 1.94 and 3.71, respectively.

Statistics

As AMP measurements were highly repeatable (interclass correlation coefficient = 0.861 for avidin and 0.953 for lysozyme), we used the average avidin and lysozyme concentration values of each biological sample for all later analyses. Moreover, as concentrations of avidin and lysozyme were not correlated (Spearman correlation, rho = 0.014, p = 0.869), we built separate models for predicting concentrations of these two AMPs, or used both AMPs as separate model predictors. Eggs were clustered into quadruplets during the experimental phase of our study, which may have affected the probability of BTSP, as shown in our previous study (Javůrková et al., 2014). To account for this source of data non-independence, quadruplet identities were included as random intercepts into all models, unless otherwise stated.

Generalised Linear Mixed Models (GLMMs) with Gaussian distributed errors were used to test whether egg volume was related to AMP concentration and whether AMP concentrations were affected by partial incubation, clutch covering with nest-lining feathers or interactions between these two variables. The effect of egg-white AMP concentration, along with effects of the above-mentioned incubation treatments, on incidence of BTSP and hatching success were analysed using logistic GLMMs (binomial error distribution, logit link). Next, using a subset of eggs positive for BTSP (i.e., number of bacterial cells estimates per 1 ml of albumen > 1) and GLMMs with Gaussian error distribution, we assessed whether intensities of bacterial penetration (log10 scaled) were predicted by concentrations of the two egg-white AMPs. Finally, we assessed whether there was any association between AMP concentration and selected hatchling morphological traits. Peig & Green (2009) showed that scaled BMI is a good indicator of the relative size of energy reserves in a homogenous population. Consequently, we used (i) residual body mass adjusted for the effect of egg volume (i.e., residuals from a linear regression on body mass vs. egg volume) and (ii) scaled body mass index (BMI), a condition index based on duckling body mass and morphometric measurements (i.e., tarsus length) as response variables, with AMP concentration, along with the effect of partial incubation (known to affect hatchling morphological traits; see Javůrková et al. (2014)), used as predictors. AMP effect on phenotype traits was modelled using linear regression since mixed models exhibited poor convergence on this data subset. Moreover, there were only two quadruplets with more than a single egg successfully hatched, suggesting a negligible effect of data non-independence on the outcomes of these analyses.

As data on avidin concentrations exhibited skewed distribution, we used log10  transformed values in all statistical calculations. Models were fitted using the R package lme4 (Bates et al., 2015) running in R software (R-Core-Team, 2020) and Rstudio version 1.1.453 (RStudioTeam, 2015). To select the best minimal adequate model (MAM), i.e., the most parsimonious model with all effects significant, backward elimination of non-significant terms in the GLMM was applied (Crawley, 2007). During this process, non-significant interactions were eliminated as first followed by non-significant main effects. Change in deviance between the model containing this term and the reduced model assuming χ2 or F distribution of difference in deviances was used for the significance assignment of a particular explanatory variable degrees of freedom were equal to the difference in degrees of freedom between the models with and without the term in question (Crawley, 2007).

Results

Effect of partial incubation and clutch covering with nest-lining feathers on egg-white AMP concentration

There was no association between egg volume and lysozyme or avidin concentrations (Δ d.f. = 1, χ2 = 0.05, p = 0.816 and χ2 = 0.33, p = 0.566, respectively; Table 1). Lysozyme concentrations was significantly higher in partially incubated eggs (Δ d.f. = 1, χ2 = 25.72, p < 0.001; Table 1, Fig. 1), while non-significant difference was observed for avidin (Δ d.f. = 1, χ2 = 3.28, p = 0.070; Table 1, Fig. 1). Clutch covering, or the interaction between clutch covering and partial incubation, had no effect on avidin and lysozyme concentration (p > 0.300 in all cases; Table 1).

Table 1 Results of GLMM evaluating egg-white antimicrobial protein concentration (Avidin and Lysozyme) as a response of partial incubation, clutch covering with nest-lining feathers and their interactions.

Step-wise elimination of nonsignificant terms was used to select the best minimal adequate model (MAM). Predictors retained in the MAM are in bold. Significance (p) was assessed based on deviance change (χ2) and corresponding degrees of freedom (Δ d.f.).

Response	Predictor	Δ d.f.	χ 2	p	
Avidin	Partial incubation	1	3.281	0.070	
	Clutch covering	1	0.000	0.987	
	Partial incubation × Clutch covering	1	0.005	0.946	
Lysozyme	Partial Incubation	1	25.716	<0.001	
	Clutch covering	1	0.755	0.385	
	Partial incubation × Clutch covering	1	0.951	0.330	

Figure 1 Variation of egg-white AMPs concentrations in Mallard (Anas platyrhynchos) eggs treated with partial incubation.

(A) Lysozyme and (B) Avidin, eggs treated with partial incubation (incub.), control un-incubated eggs (unincub.). Also shown are GLMM-based probability values.

Effect of egg-white AMP concentration on BTSP

We detected BTSP in 91 of 160 experimental eggs (57%), with a mean BTSP intensity of 3.4 × 105 bacterial cells per 1 mL of egg white (range: 102–104 bacterial cells per 1 mL; see Fig. S2). Concentration of egg-white lysozyme and avidin had no effect on the incidence of BTSP (Δ d.f. = 1, χ2 = 0.05, p = 0.82 and Δ d.f. = 1, χ2 = 0.01, p = 0.999, respectively; Table 2), and incidence of BTSP was unaffected by the interaction between AMP concentration and experimental treatment (p > 0.2 in all cases; Table 2). There was also no correlation between intensity of BTSP and egg-white lysozyme (Δ d.f. = 1, χ2 = 0.42, p = 0.517; Table 2) or avidin (Δ d.f. = 1, χ2 = 2.64, p = 0.104; Table 2) concentrations in a subset of penetrated eggs.

Table 2 Results of GLMM evaluating variation in incidence and intensity of bacterial trans-shell penetration (BTSP).

Egg-white lysozyme and avidin concentrations, partial incubation, clutch covering with nestlining feathers, and their interactions were used as predictors. Predictors retained in the minimal adequate model after step-wise elimination of nonsignificant variables are in bold. Also shown are probability values (p), χ2 values and associated degrees of freedom (Δ d.f.).

Response	Predictor	Δ d.f.	χ 2	p	
BTSP Incidence	Lysozyme	1	1.619	0.203	
	Avidin	1	0.001	0.981	
	Partial incubation	1	0.052	0.820	
	Clutch covering	1	0.052	0.820	
	Lysozyme × Partial incubation	1	0.209	0.648	
	Lysozyme × Clutch covering	1	0.488	0.485	
	Avidin × Partial incubation	1	0.660	0.417	
	Avidin × Clutch covering	1	0.328	0.567	
BTSP Intensity	Incubation	1	3.071	0.080	
	Avidin	1	2.642	0.104	
	Lysozyme	1	0.420	0.517	
	Clutch covering	1	0.420	0.517	
	Lysozyme × Partial incubation	1	1.480	0.224	
	Lysozyme × Clutch covering	1	0.772	0.379	
	Avidin × Partial incubation	1	0.037	0.847	
	Avidin × Clutch covering	1	0.305	0.581	

Effect of egg-white AMP concentration on hatching success

In the present study, we found that higher hatching success of partially incubated eggs was unaffected by egg volume (GLMM with binary response: Δ d.f. = 1, χ2 = 0.056, p = 0.813), or egg-white lysozyme and avidin concentrations (Δ d.f. = 1, χ2 = 1.26, p = 0.262 and Δ d.f. = 1, χ2 = 1.58, p = 0.209, respectively; Table 3). Similarly, we found no support for any interaction between both AMP concentrations and experimental treatments on hatching success (p > 0.1 in all cases, Table 3).

Table 3 Results of GLMM evaluating variation in hatching success.

Egg-white lysozyme and avidin concentrations, partial incubation, clutch covering with nestlining feathers and their interactions were used as predictors. Predictors retained in the minimal adequate model after step-wise elimination of nonsignificant variables are in bold. Also shown are probability values (p), χ2 values and associated degrees of freedom (Δ d.f.).

Predictor	Δ d.f.	χ 2	p	
Partial incubation	1	8.796	0.003	
Lysozyme	1	1.257	0.262	
Avidin	1	1.575	0.209	
Clutch covering	1	1.575	0.209	
Lysozyme × Partial incubation	1	0.645	0.422	
Lysozyme × Clutch covering	1	2.579	0.108	
Avidin × Partial incubation	1	0.080	0.778	
Avidin × Clutch covering	1	0.844	0.358	

Effect of egg-white AMP concentration on hatchling phenotype

Partially incubated eggs produced hatchlings with significantly reduced residual body mass and scaled BMI (F(1,25) = 23.98, p < 0.001 and F(1,24) = 10.97, p = 0.002, respectively; Table 4). When accounting for this source of variation, we observed no effect of AMP concentration on residual body mass (F(1,24) = 0.02, p = 0.903 for lysozyme and F(1,24) = 0.01, p = 0.935 for avidin; Table 4). At the same time, avidin failed to predict variation in scaled BMI (F(1,23) = 0.14, p = 0.711; Table 4); however, scaled BMI was significantly reduced in hatchlings originated from eggs with higher concentrations of egg white lysozyme (F(1,24) = 7.23, p = 0.013) after statistical control for the variation induced by partial incubation (Table 4, Fig. 2).

Table 4 Results of GLMM evaluating variation in morphometric parameters of hatchlings.

Variability in Residual body mass (i.e. body mass adjusted for egg volume), and Scaled BMI (condition index based on body mass and tarsus length) due to the effect of egg-white lysozyme and avidin concentrations and partial incubation was evaluated. Data were analysed using linear models assuming Gaussian distribution of residuals. Predictors retained in the minimal adequate model after step-wise elimination of nonsignificant variables are in bold. Also shown are probability values (p), F-statistic values (F) and associated degrees of freedom (d.f.).

Response	Predictor	d.f.	F	p	
Residual body mass	Partial incubation	(1,26)	23.982	0.000	
	Lysozyme	(1,25)	0.015	0.903	
	Avidin	(1,24)	0.007	0.935	
Scaled BMI	Lysozyme	(1,25)	10.965	0.003	
	Partial incubation	(1,25)	7.227	0.013	
	Avidin	(1,24)	0.140	0.711	

Figure 2 Effect of egg-white lysozyme concentrations on scaled BMI of Mallard (Anas platyrhynchos) hatchlings.

Regressions were adjusted for the effect of partial incubation treatment. Also shown are predictions and 95% confidence intervals.

Discussion

In the present study, partial incubation of mallard eggs caused alterations in the concentration of egg-white AMPs, with lysozyme showing a significantly higher concentration in partially incubated eggs. This concurs with a recent study that compared the proteomic profile of fertile chicken eggs during the first 12 days of incubation, demonstrating a substantially higher egg-white lysozyme content in fertilised eggs at day 12 compared with non-incubated eggs at day 0 (Zhu et al., 2019). Similarly, Guyot et al. (2016b) observed a gradual increase in egg-white protein concentration from day 0 to 12 of incubation in fertilised chicken eggs. It is important to note here that additional egg-white protein synthesis is impossible after oviposition; hence, the observed higher egg-white protein concentration in partially incubated eggs is most probably the result of a substantial loss of water from the egg white during partial incubation due to embryo growth, the synthesis of embryonic membranes and extraembryonic fluids, and evaporation (Guyot et al., 2016b; Romanoff & Romanoff, 1933). In our previous experimental study, however, partial incubation had a non-significant effect on egg-white lysozyme concentrations in quail (Coturnix japonica) and pigeon (Columba livia domestica) eggs (Svobodová et al, 2019), while other studies have documented either a slight decrease in egg-white lysozyme in chicken eggs during the early phase of full incubation (Fang et al., 2012a; Fang et al., 2012b), or a decrease in egg-white lysozyme concentration in precocial chicken eggs (Cunningham, 1974) and altricial red-capped lark eggs (Grizard et al., 2015) following full incubation. Clearly, there is inconsistency as regards temperature-induced changes in egg-white AMPs under different incubation modes. While this might suggest that different embryonic developmental stages are playing a role (see (Guyot et al., 2016b)), other studies have suggested that egg-white lysozyme decreases due to protein aggregation (Liu, Qiu & Ma, 2015; Qiu et al., 2012), lysozyme binding to other proteins (Kato, Imoto & Yagishita, 1975) or lysozyme degradation soon after incubation (Fang et al., 2012a). Recent works, however, have suggested that thermal aggregation of proteins, and the resulting changes in protein abundance, are highly dependent on the content of particular amino acids, such as arginine, lysine or aspartic acid, which act as protein stabilisers and/or destabilisers (Anumalla & Prabhu, 2019; Hong et al., 2017). Further, temperature-induced levels of protein aggregation have been shown to be linked with the concentration of other heat-sensitive egg-white proteins, such as ovotransferrin (Iwashita, Handa & Shiraki, 2019). As proteomic and amino acid profiles vary considerably between species (Shawkey et al., 2008; Sun et al., 2017), it is highly likely that, in addition to changes attributable to water loss (Guyot et al., 2016b; Romanoff & Romanoff, 1933), observed differences in egg-white lysozyme concentration could be the result of differing ratios and concentrations of aggregation-preventing arginine and/or ovotransferrin in the mallard eggs used in our study. Unfortunately, we were unable to test for such associations in this study as the results for replicate measurements of egg-white ovotransferrin concentration were highly variable. While further experimental testing is needed to prove the relationship with aggregation-preventing egg-white substances, we suggest that the potential for such effects should be considered during future research focused on the thermal properties of egg-white.

In our study, we found no support for the role of egg-white AMP concentration in preventing bacterial trans-shell penetration in vivo, with neither lysozyme nor avidin concentration affecting the incidence or intensity of BTSP. This was still true following the significantly higher concentration of egg-white lysozyme in partially incubated eggs. This finding is supported by a recent experimental study of Baron et al. (2020) testing the effect of egg-white proteins (egg-white fraction >10 kDa) on egg-white anti-Salmonella activity, demonstrating that presence of egg-white proteins played a minor role in the bactericidal activity of egg white at 45 °C, suggesting that egg-white low-mass components (<10 kDa) have a greater impact on temperature-induced bactericidal activity of chicken egg white at 45 °C. On the contrary, in our previous experimental study, egg whites enriched in ovo with hen egg-white lysozyme significantly increased their in vitro antimicrobial action against indicator strains (Svobodová et al, 2019). To date, however, there have been no studies evaluating the in vivo antimicrobial potential of naturally varying egg-white AMPs or associated trans-shell microbial penetrations. In our previous study, we also noted selective in vitro antimicrobial activity in egg whites from precocial eggs treated with partial incubation, with enhanced proliferation of a beneficial probiotic bacterial strain (Svobodová et al, 2019). Similarly, incubation was shown to shift eggshell microbiota diversity from initially diverse communities that included opportunistic pathogens toward less diverse communities with less harmful, or even beneficial, microorganisms dominating (Grizard et al., 2014; Lee et al., 2014). It would appear, therefore, that partial incubation as a mechanism acts outside the egg to modulate eggshell microbial communities toward harmless or beneficial microorganisms, and inside the egg to maintain beneficial bacterial invaders. Moreover, the protective roles of incubation and egg-white AMPs against pathogenic microorganisms appears to be most effective during the early phase of embryonic development, while developing extra-embryonic structures appear to play a greater protective role later in the incubation process (Guyot et al., 2016b; Hincke et al., 2019). Clearly, therefore, the role of egg-white AMPs in modulating microbial trans-shell invaders during different incubation phases is complex, and more in vivo studies will be needed to fully understand the mechanisms behind this complexity.

Previous studies have found that egg-white lysozyme has a beneficial maternal effect when deposited into the eggs by the female of precocial (Bonisoli-Alquati et al., 2010; Cucco et al., 2007; Kozuszek et al., 2009) and altricial birds (Boonyarittichaikij et al., 2018), resulting in a higher egg hatching rate (Cucco et al., 2007) or improved hatchability and immunocompetence of nestlings (Saino et al., 2002). However, our results indicate that higher egg-white lysozyme levels may also had a negative impact on hatchling body condition (expressed as scaled BMI). This is in accordance with our previous study, where an experimental increase in egg-white lysozyme in precocial quail eggs resulted in a reduced tarsus length in hatchlings (Javůrková et al., 2015). While the mechanism of action is not yet clear, lysozyme is known to play a growth-regulating role in the development of embryonic cartilage and skeletal structures (Kuettner et al., 1970), including inhibition of mouse bone collagenase activity, which could significantly affect development of particular skeletal elements (Sakamoto et al., 1974).

In this study, egg-white avidin concentration appeared to have no impact on mallard hatchling phenotype. While we previously documented egg-white avidin as altering chick phenotype in quail (Krkavcová et al., 2018), the growth-inhibition effect in this case was strongly dependent on egg weight, since only those chicks originating from lighter eggs enriched with avidin in ovo had a reduced tarsus length. It follows, therefore, that while egg-white AMPs may fulfil a protective antimicrobial role for the embryo during the early phases of embryo development, increases in their concentration may significantly compromise embryo growth and negatively affect hatchling morphological traits and condition in precocial birds.

As in the case of partial incubation, we failed to find any effect of clutch covering with nest-lining feathers on egg-white AMPs or incidence and intensity of bacterial trans-shell penetration. Though clutch covering has been shown to insulate eggs against ambient temperatures (Pinowski et al., 2006; Prokop & Trnka, 2011), it would appear that its main purpose is to maintain eggs at optimal temperatures around physiological zero, thereby sustaining egg viability and improving hatchability and hatchling growth performance (Dawson, O’Brien & Mlynowski, 2011; Peralta-Sanchez, Moller & Soler, 2011; Stephenson, Hannon & Proctor, 2009), rather than to alter temperatures to levels that lead to changes in the egg-white AMP profile. Just two studies have shown that nest material and nest-lining feathers have a strong antimicrobial effect, both indicating an ability to shift eggshell microbiota in hoopoe (Upupa epops) (Ruiz-Castellano et al., 2019; Ruiz-Castellano et al., 2016). As evidence for the antimicrobial action of nest-lining feathers is lacking in other bird species, and we observed no effect of alteration of egg-white AMPs on bacterial trans-shell penetration, it is highly likely that nest-lining feathers in our study species may only have an antimicrobial effect on the eggshell itself, without affecting the antimicrobial potential of egg-white; alternatively, its primary function may be clutch insulation and/or protection against visually-oriented predators (Kreisinger & Albrecht, 2008).

Conclusions

We were able to show that partial incubation, a behavioural mechanism having a range of functions, from antipredator nest protection to maintaining egg viability, also has the ability to alter concentrations of particular egg-white AMPs during the pre-incubation phase. Furthermore, while concentrations of particular egg-white AMPs were not associated with reduced intensity and incidence of bacterial trans-shell penetration in eggs of a precocial bird in vivo, increased concentration of egg-white lysozyme may play a growth-modulating role during embryogenesis, at least in our precocial model species. While the growth-modulating role of particular egg-white AMPs during the various developmental stages of avian embryos requires further testing, our results are some of the first to point out these potential relationships.

Supplemental Information

Supplemental Information 1 Overview scheme of experimental design

Experimental eggs (n = 160) were exposed in semi-artificial nests (n = 40) for 9 days in a natural breeding habitat. I-INCUB = partially incubated eggs, I-UNINCUB = un-incubated eggs. Note: Figure was adopted from the previous study of Javůrková et al. (2014): Ibis 156, 374–386. Copyright 2014 by Veronika Javůrková.

Click here for additional data file.

Supplemental Information 2 Probability and intensity of bacterial trans-shell penetration (BTSP)

Proportion (%) of penetrated (black bar) vs. non-penetrated (white bar) experimental eggs and intensities of BTSP in penetrated eggs (grey bars) expressed as the number of bacterial cells per one mL of egg white. Note: figure was adopted from the previous study of Javůrková et al. (2014): Ibis 156, 374–386. Copyright 2014 by Veronika Javůrková.

Click here for additional data file.

Supplemental Information 3 Daily exposure (hours) of partially incubated experimental eggs

Eggs were exposed in an incubator OvaEasy 190 Advance (Brinsea Products Inc., Titusville, FL, USA) at 37.6 °C with a relative humidity of 60% for a period of 9 days. Note: table was adopted from the previous study of Javůrková et al. (2014): Ibis 156, 374–386. Copyright 2014 by Veronika Javůrková.

Click here for additional data file.

We thank Eva Krkavcová for her help in the laboratory and for the AMP concentrations analysis, and Kevin Roche for professional English proofreading.

Additional Information and Declarations

Competing Interests

Author Contributions

Animal Ethics

Data Availability

The authors declare there are no competing interests.

Jana Svobodová analyzed the data, authored or reviewed drafts of the paper, and approved the final draft.

Jakub Kreisinger conceived and designed the experiments, performed the experiments, analyzed the data, prepared figures and/or tables, authored or reviewed drafts of the paper, and approved the final draft.

Veronika Gvoždíková Javůrková conceived and designed the experiments, performed the experiments, analyzed the data, authored or reviewed drafts of the paper, and approved the final draft.

The following information was supplied relating to ethical approvals (i.e., approving body and any reference numbers):

All experiments and analyses were performed in accordance with relevant institutional guidelines and regulations. The experiment was carried out under permission No. 63479/2016-MZE-17214, issued by the Ministry of Agriculture on behalf of the Government of the Czech Republic.

The following information was supplied regarding data availability:

The dataset generated and analysed in this study is available at figshare: Gvozdikova Javurkova, Veronika (2021): raw data_Mallard_proteins_BTSI_morpho.txt. figshare. Dataset. https://doi.org/10.6084/m9.figshare.14554203.v1.

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
