# Peer review of "Temperature-induced changes in egg white antimicrobial concentrations during pre-incubation do not influence bacterial trans-shell penetration but do affect hatchling phenotype in Mallards"

_PeerJ, doi:10.7717/peerj.12401_

## Round 0.1 · original submission · Major Revisions

Please correct and submit to a professional English correction service.

·

Basic reporting

no comment

Experimental design

there are fundamental errrors in this study. The authors state that they measured lysozyme concentration, but in reality they measure antibacterial activity against one Micrococcus strain. It is well known that, in the period after the egg was laid, the pH increases and this triggers spontaneous degradation of ovotransferrin into a range fo antimicrobial peptides. This may explain why the authors think they have seen an increase in lysozyme concentration, which actually is biologically impossible. The authors state that they have measured avidin concentration, but nowhere in their M&M they mention the use of biotin, which is the essential key of the test of Gan & Marquardt and Shawkey et al.
The authors state they measured infection of egg white. however, they only quantified bacterial DNA, which very well may come from dead bacterial cells, killed by the antimicrobial components of the egg white, so this is also not correct.

Validity of the findings

Underlying data have not been provided: we miss the data on incidence and intensity of BTSI (how many eggs were positive?).
Lysozyme in produced in the magnum of the oviduct in the female, and deposited in the egg white in this way. Thus it is biologically impossible that the lysozyme concentration would increase after oviposition.
The authors should check the definition of the word 'infection'.
Hatching success increases with partial incubation of eggs. There is a good chance that this is the consequence of survival of the smaller embryo's, which would not have survived otherwise.
There is no proof of a causal relationship between lysozyme concentration and scaled body mass index, even if lysozyme concentration would have been measured.

Reviewer 2 ·

Basic reporting

Clear, unambiguous, professional English language used throughout: YES
Intro & background to show context. Literature well referenced & relevant: YES, I have added some literature citations in the specific suggestions below.
Structure conforms to PeerJ standards, discipline norm, or improved for clarity.
Figures are relevant, high quality, well labelled & described.: YES

Experimental design

Original primary research within Scope of the journal: YES
Research question well defined, relevant & meaningful. It is stated how the research fills an identified knowledge gap: YES
Rigorous investigation performed to a high technical & ethical standard: YES
Methods described with sufficient detail & information to replicate: YES

Validity of the findings

All underlying data have been provided; they are robust, statistically sound, & controlled.: YES

Additional comments

I have added some suggestions and literature citations in the attached PDF file

Annotated reviews are not available for download in order to protect the identity of reviewers who chose to remain anonymous.

---

## Round 0.2 · Minor Revisions

Reviewer 2 has noted that the point about Lysozyme is not completely addressed in the revised submission. Lysozyme is mainly produced in the oviduct of the female, and deposited in the egg white. Thus it is not clear how lysozyme concentration increases after oviposition. The authors need to discuss this in the manuscript, for the sake of transparency and not just include comments in a response to reviewers document. It is essential to explain and discuss why authors use lysozyme as the dependent variable here

Reviewer 2 ·

Basic reporting

no comment

Experimental design

no comment

Validity of the findings

no comment

Additional comments

Typo checks:
line 43: alterate
line 232: affinity

Reviewer 3 ·

Basic reporting

The Ms is well written except in the abstract and at the beginning of the indtroduction English imporvemnts may be necessary.

The MS is well organized and all the relevant results shown. All tables and figures are necessary and clearly represented.

Literature is fully explored.

However, I have a problem with hypotheses and predictions. This is related to a comment mentioned by another reviewer earlier and which to my opinion is not fully answered (see later).

Experimental design

Most of the methods described are sufficient. However I miss, as far as I can see, an explanation of the phenotypic measurements in the methods section e.g. there is no explanation, information about scaled BMI or residual body mass, e.g. when and how they were measured and what is meant with it, etc.!
Another point is whether phenotype is a proper term alternatively you may say what it is namely individual quality or condition. Phenotype frequently also implies other characteristics!

Validity of the findings

The authors produced already similar studies with other species, thus the novelty of the study is limited, but the topic is very interesting and there is not much known yet.
This study is mixed and contains experimental and correlational parts. Thus conclusions should be still drawn with care.

Additional comments

In this study the authors experimentally examine the importance of two parent behaviours, namely partial incubation and covering the eggs during the egg laying period. Both behaviours may possible influence egg temperature and consequently egg shell microbiome and the concentration of egg white lysozime, avidin and consequently the amount of bacteria penetrating through the eggshell. Finally they investigate, wether there is an effect on hatching success and offspring condition. As a model species they use the mallard, a precocial bird species. According to the results, from the two behaviours, partial incubation seems to effect offspring condition whereas covering the eggs seems not important.

I have one major point, which seems to be raised already by another author ealier. I do not have the feeling that this point is answered satisfactory, at least not in the MS.
As the reviewer pointed out Lysozyme is mainly produced in the the oviduct of the female, and deposited in the egg white. Thus it is impossible that the lysozyme concentration increases after oviposition.
The authors need to deal with that point in more detail in the MS and not only in the rebuttal letter!
It is essential to explain and dicuss why you use lyozyme as the dependent variable here!!

In this context the introduction is well written and covers most of the important areas but one topic the authors may inlcude (introduction and discussion) is maternal investment i.p. related to offspring immunity.

Abstract: English seems poor.
The first paragraph in the introduction is rather inflated and in principle hard to understand, you may consider to simplify and reduce that part.

Statistical procedures and results are well done.

---

## Round 0.3 · accepted · Accept

Congratulations, your manuscript has been approved.

Reviewer 3 ·

Basic reporting

The MS has significantly improved after the revision.

Experimental design

no comment

Validity of the findings

no comment

Additional comments

One thing which came to me and you may mention in the MS (introduction discussion) is, that in other species, also other material than feathers is used to cover eggs (e.g. starlings), and these nest material (aromatic plants) can have strong antimicrobial properties.
This may open up new future research possibilities!